# METAFOOD3D: 3D FOOD DATASET WITH NUTRITION VALUES

## ABSTRACT

Food computing is both important and challenging in computer vision (CV). It significantly contributes to the development of CV algorithms due to its frequent presence in datasets across various applications, ranging from classification and instance segmentation to 3D reconstruction. The polymorphic shapes and textures of food, coupled with high variation in forms and vast multimodal information, including language descriptions and nutritional data, make food computing a complex and demanding task for modern CV algorithms. 3D food modeling is a new frontier for addressing food related problems, due to its inherent capability to deal with random camera views and its straightforward representation for calculating food portion size. However, the primary hurdle in the development of algorithms for food object analysis is the lack of nutrition values in existing 3D datasets. Moreover, in the broader field of 3D research, there is a critical need for domain-specific test datasets. To bridge the gap between general 3D vision and food computing research, we introduce MetaFood3D. This dataset consists of 637 meticulously scanned and labeled 3D food objects across 108 categories, featuring detailed nutrition information, weight, and food codes linked to a comprehensive nutrition database. Our MetaFood3D dataset emphasizes intra-class diversity and includes rich modalities such as textured mesh files, RGB-D videos, and segmentation masks. Experimental results demonstrate our dataset's significant potential for improving algorithm performance, highlight the challenging gap between video captures and 3D scanned data, and showcase the strengths of MetaFood3D in high-quality data generation, simulation, and augmentation.

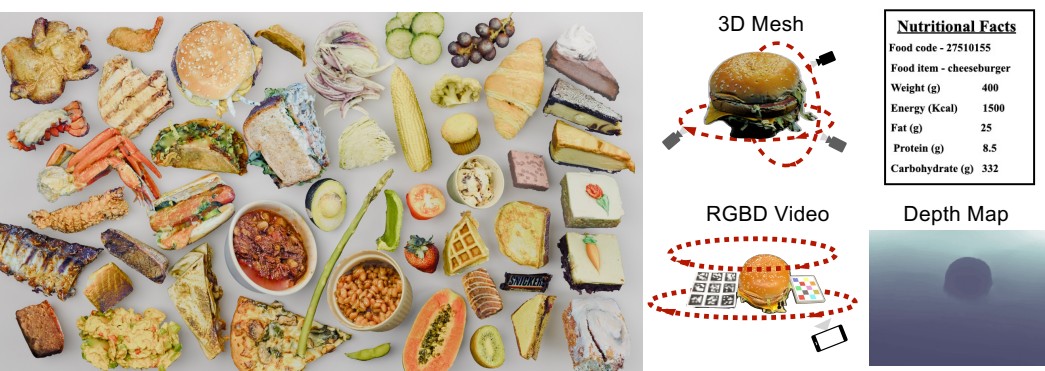

Figure 1: **MetaFood3D is a real-scan 3D food dataset featuring diverse ready-to-eat 3D textured meshes, 720-degree RGBD video captures, and rich nutrition value annotations.**

## 1 INTRODUCTION

Food is fundamental to our existence, serving not just as a basic necessity for survival but also as a crucial aspect of our social interactions, where sharing images, videos, and even virtual food experiences in video games is commonplace. Food-related image analysis is crucial for monitoring and improving dietary habits across different age groups, as it enables personalized nutrition interventions, supports early detection of dietary deficiencies, and promotes healthier lifestyles tailored to the specific needs of children, adults, and the elderly. In the field of computer vision, food has played a significant role

in advancing algorithms, given its frequent occurrence in both specialized and general datasets for tasks such as classification Gao et al. (2022a); He & Zhu (2021); Jiang et al. (2019); Raghavan et al. (2024), instance segmentation Lan et al. (2023), and 3D object reconstruction Qian et al. (2023).

Food data is uniquely complex due to unbalanced classes, intricate textures, hierarchical categorization, and ambiguous shapes. Often, food images are taken from close distances, with varying camera angles leading to diverse visual representations. Typical single-view-image depictions fall short of providing comprehensive views, obscuring critical details about ingredients and portions. *E.g.*, an overhead image of a sandwich might display only the bun, while a side view could expose the bun, meat, and toppings in greater detail, highlighting the limitations of single-view image analysis.

Accurate measurement is crucial for various food-related tasks, especially under the context of precise dietary assessment, which can serve as a valuable digital biomarker, offering a quantitative and objective measure of an individual's nutritional intake and its potential impact on their health status. A significant challenge in dietary assessment is to accurately estimate portion sizes from food images Tahir & Loo (2021b). Various approaches have been developed to tackle this problem, including image based regression Thames et al. (2021), regression on segmentation masks He et al. (2013); Konstantakopoulos et al. (2023), mapping to handcrafted 3D shape templates Jia et al. (2023), 3D reconstruction from multiple images Konstantakopoulos et al. (2021), and utilizing depth information Graikos et al. (2020). However, the lack of 3D information for individual food object leads to inaccuracies and challenges in generalization. Even with depth data, accurately representing empty spaces beneath food objects remains a challenge, as foods on a plate can exhibit a wide range of 6D poses and stacking relationships.

Recent advancements in 3D vision algorithms, particularly in novel view synthesis Mildenhall et al. (2021), surface reconstruction Wang et al. (2021), and 3D object generation Lin et al. (2023), indicate a promising direction for overcoming these issues. Utilizing 3D methodologies in food-related research offers inherent advantages, such as mitigating challenges posed by varied camera views through novel view synthesis or rendering from learned geometries. These approaches can facilitate the direct computation of food volume per food item for dietary studies, making the process more precise, straightforward, and explainable compared to existing methods. However, at this stage, the main obstacle to applying these 3D algorithms to food-related tasks is the lack of well constructed food datasets.

Many generic large-scale 3D datasets Deitke et al. (2023); Wu et al. (2023); Francis et al. (2022) have recently been released, fueling the development of 3D vision algorithms Shi et al. (2023); Liu et al. (2023b). Yet, there is a notable scarcity of food-specific datasets to train and evaluate 3D algorithms on food-related tasks. Existing 3D datasets with food generally lack dietary annotations such as weight, calories, and other nutrition values, which is crucial for developing 3D or image-based dietary assessment algorithms. Furthermore, there is a shortage of benchmark 3D food datasets featuring diverse intra-class variation. For instance, the OmniObject3D dataset Wu et al. (2023) includes 2,837 food objects, but the selection of its food instances fails to emphasize the appearance variations within each food category. Many food items in OmniObject3D, such as lemons, exhibit similar appearances and geometries within the same category.

To bridge the gap between general 3D vision and food computing, and to provide a unique benchmark for both general and food-specific downstream tasks, our dataset MetaFood3D (as shown in Figure 1) endeavors to develop a food-specific 3D dataset that advances dietary analysis from 2D to 3D. MetaFood3D includes a total of 637 3D food objects in 108 food categories. Each food object in the dataset is meticulously labeled with detailed nutrition information, weight, and food codes linked to a comprehensive nutrition database Montville et al. (2013). We emphasize intra-class diversity by collecting foods with varying appearances and nutritional information. Beyond nutritional facts, our dataset includes rich modalities such as textured mesh files, RGB-D videos, and segmentation masks. Additionally, the dataset incorporates hierarchical relationships characterized by specifying sub-food-categories, known as food items, within general food categories, facilitating tasks related to fine-grained classification. Finally, we establish baselines for nutrition estimation, perception, reconstruction, and generation tasks. Our experiments demonstrate that our dataset has significant potential for improving performance and highlight the challenging gap between video captures and 3D scanned data. Furthermore, we show the potential of our dataset for high-quality data generation, simulation, and augmentation by presenting high-quality visual results.

| | Multiview/video | Depth | Inst Mask | Mesh | Size Calibration | Nutrition | Food categories | Samples |
|---|---|---|---|---|---|---|---|---|
| *Food Specific Datasets* | | | | | | | | |
| Food2K Min et al. (2023) (2D) | | | ✓ | | | | 2,000 | 1 Million |
| ECUSTFD Liang & Li (2017) (2D) | | | | | ✓ | ✓ | 19 | 2,978 |
| Nutrition5K Thames et al. (2021) (2D) | ✓ | ✓ | | | ✓ | ✓ | 250 | 5,006 |
| NutritionVerse3D Tai et al. (2023) | ✓ | | ✓ | ✓ | | ✓ | 54 | 105 |
| *Generic 3D Datasets* | | | | | | | | |
| GSO Francis et al. (2022) | | | ✓ | ✓ | ✓ | | 0 | 0 |
| CO3D Reizenstein et al. (2021a) | ✓ | ✓ | ✓ | | ✓ | | 10 | 5,077 |
| OmniObject3D Wu et al. (2023) | ✓ | | ✓ | ✓ | ✓ | | 85 | 2,837 |
| Ours | ✓ | ✓ | ✓ | ✓ | ✓ | ✓ | **108** | 637 |

Table 1: **Public Datasets with Real-world Food Objects.** "Samples" represents the total number of food data samples in the dataset. Note that we exclude food toys in GSO.

## 2 RELATED WORK

In this section, we provide detailed reviews of related food and 3D object datasets and a brief review of relevant downstream tasks. The features of these datasets are summarized in Table 1.

**Food Datasets** are primarily developed to answer key questions in food computing: "What is the food in the image?", "What is the portion size?", and "What is the nutritional content of the food?". While numerous food classification datasets exist, ranging from the classic Food-101 dataset Bossard et al. (2014) to the latest Food2K dataset Min et al. (2023), datasets for portion estimation or macro-nutrient estimation are significantly fewer. This scarcity is due to the complexity and labor-intensiveness of collecting multi-modal data with physical food object references. Numerous efforts have been undertaken to mitigate the need for gathering data on physical objects. These include leveraging images and metadata from recipe websites Ruede et al. (2021) or creating synthetic data by pasting image textures onto predefined geometries Yang et al. (2021). However, these approaches have fundamental flaws, as the relationship between the food appearance and the food weight is not validated by real food items. Despite various proposals for ground-referenced food portion estimation datasets in existing literature Lo et al. (2020); Tahir & Loo (2021a); Wang et al. (2022); Konstantakopoulos et al. (2024), only three datasets that include nutrition values are publicly available: ECUSTFD Liang & Li (2017), Nutrition5K Thames et al. (2021), and NutritionVerse3D Tai et al. (2023). The ECUSTFD dataset contains no geometry information. In the Nutrition5K dataset, food items are mixed together without segmentation masks, making it infeasible to perform nutrition and geometric modeling for individual food items. The NutritionVerse3D dataset, which includes models from FoodVerse Tai et al. (2022), is small-scale, containing 105 3D food models across 42 unique food types. The food items are not calibrated in size and the selection of food types appears to be random and imbalanced.

**3D Object Datasets** focus either on synthetic objects created by humans or on real-world objects that are manually scanned. Synthetic object datasets, such as ShapeNet Chang et al. (2015) and Objaverse Deitke et al. (2023), are unsuitable for dietary assessment applications due to their artistic object appearances and non-referenced scales. Real-world scanned objects offer realistic appearances and geometry, but many real-world 3D object datasets primarily focus on non-perishable commercial household items, including Google Scanned Objects (GSO) Francis et al. (2022), CO3D Reizenstein et al. (2021a), YCB Objects Calli et al. (2015), AKB-48 Liu et al. (2022), and MetaGraspNetV2 Gilles et al. (2023). Some real-world scanned object datasets do include food items, but they often suffer from limitations such as a small number of food categories Reizenstein et al. (2021a). Additionally, the selection of food items is often random and does not reflect the distribution of commonly eaten foods, leading to bias in dietary assessment Wu et al. (2023).

**Food Data Analysis for Dietary Assessment:** Existing food portion and nutrition value estimation methods can be classified into four main categories: stereo-based Puri et al. (2009); Dehais et al. (2017), depth-based Lo et al. (2019); Fang et al. (2016), model-based Xu et al. (2013); Jia et al. (2014), and neural network-based methods He et al. (2020); Shao et al. (2021); Ma et al. (2023); Vinod et al. (2022); He et al. (2021); Thames et al. (2021); Shao et al. (2023). Recently, a 3D model-based method Vinod et al. (2024) has demonstrated the importance of 3D models in food portion estimation by outperforming many existing methods.

**3D Point Cloud Perception:** This task seeks to classify point cloud data composed of a set of 3D coordinates. PointNet Qi et al. (2017a) was first proposed to directly process unordered raw

point cloud sets. PointNet then led to the development of new models Qi et al. (2017b); Wang et al. (2019); Xu et al. (2021a); Ma et al. (2022). Due to the characteristics of real-world point cloud data, robustness is crucial in 3D point cloud perception. Previous works Ahmadyan et al. (2021); Reizenstein et al. (2021b); Ren et al. (2022); Taghanaki et al. (2020) have studied the robustness of models on point cloud data from different domains and standardized corrupted dataset.

**Novel View Synthesis and 3D Mesh Reconstruction:** Novel view synthesis aims to generate high-quality images from new perspectives given only a few training images. Neural Radiance Fields (NeRF) Mildenhall et al. (2021) addresses this problem by training a multilayer perceptron (MLP) network to predict the color values and densities of locations in space. Recent advancements have tackled issues related to aliasing, quality, and efficiency Barron et al. (2021); Müller et al. (2022); Kerbl et al. (2023); Tancik et al. (2023). 3D mesh reconstruction aims to recreate the mesh of an object. Traditional methods like Structure from Motion (SfM) Schönberger & Frahm (2016) achieve this by determining the camera pose associated with each image. Recent approaches leverage the success of volume rendering in novel view synthesis Wang et al. (2021); Li et al. (2023); Huang et al. (2024) or employ Neural Signed Distance Fields Munkberg et al. (2022).

**3D Generation:** With advancements in novel view synthesis and generative models Rombach et al. (2022), numerous text-to-3D generation methods have emerged in the past year Liu et al. (2024). A typical pipeline involves leveraging diffusion models to generate multi-view images of an object, which are then utilized in 3D reconstruction methods to create the 3D model Shi et al. (2023); Long et al. (2023). Other approaches focus on learning Neural Signed Distance Fields to achieve 3D generation Gao et al. (2022b) .

## 3  DATASET

The selection of food objects and their multimodal labels in the MetaFood3D dataset is designed to support dietary assessment applications, which involves identifying various foods in images and estimating portion sizes and nutritional values using RGB and/or depth sensors from diverse camera angles. To accurately reflect these use cases, we first carefully selected food items and their variations based on real-world food consumption patterns, as detailed in the **Food Objects Selection** paragraph. Second, we curated the modalities and labels to capture the relevant characteristics of real-world dietary assessment data, as described in the **Data Collection** and **Annotation** paragraph. Figure 2 provides an overview of MetaFood3D, illustrating the distribution of data and energy content across food objects, as well as the intra-class variance of the collected food objects.

**Food Objects Selection:** Identifying which food objects to collect is challenging due to the vast number of food categories and the significant appearance variations even within the same category. For example, apples could be broadly categorized as fruit, but they also come in different varieties, colors, shapes, and sizes, and can be used in diverse preparations like apple pies. Determining the appropriate level of class granularity poses another challenge—should we classify broadly as "fruit," more specifically as "apple," or even further as "Fuji apple"? To address these challenges, we consulted nutrition experts and referenced an established food list from the VIPER-FoodNet (VFN) dataset Mao et al. (2021). The VFN dataset, derived from the What We Eat in America (WWEIA) database[1], provides a comprehensive overview of the American diet. It has been widely used in food computing tasks, such as long-tailed learning He et al. (2023), continual learning Raghavan et al. (2024), personalized classification Pan et al. (2023), and multimodal learning Pan et al. (2024). To enhance categorical diversity, we expanded the original 74 food categories from the VFN dataset by incorporating 34 additional categories based on data from the National Health and Nutrition Examination Survey (NHANES) Lin et al. (2022), resulting in a total of 108 food categories in the MetaFood3D dataset. One key enhancement of our dataset over the VFN dataset is the increased granularity of food code matching. While VFN matches each food category with a single general 8-digit food code from the Food and Nutrient Database for Dietary Studies (FNDDS) Montville et al. (2013), we assign each food object a specific FNDDS food code. For example, within the "Pie" category, we include specific items like "Pie, chocolate cream," "Pie, pecan," "Pie, apple," and "Pie, lemon," each with their respective FNDDS codes. This detailed matching allows for a more accurate representation of diverse food items, acknowledging their unique ingredients and nutritional profiles. By providing this level of detail, our 3D food dataset enables more precise dietary analysis and the development of sophisticated computer vision algorithms capable of distinguishing between different

---

[1]https://data.nal.usda.gov/dataset/what-we-eat-america-wweia-database

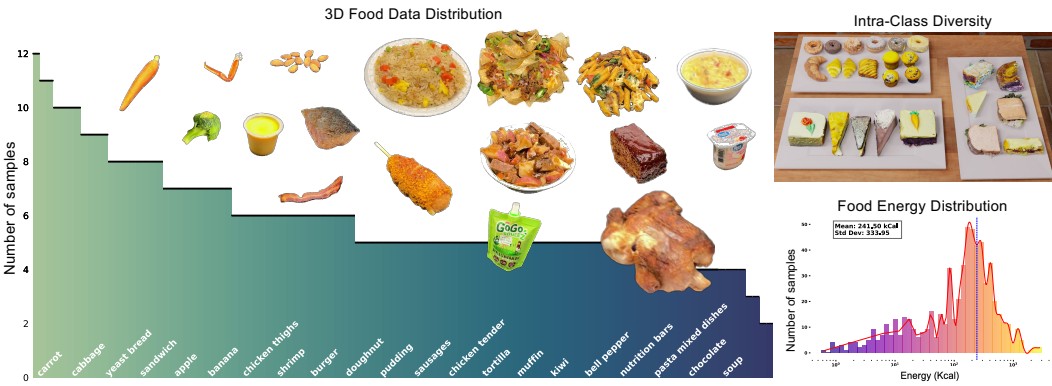

Figure 2: **The distribution of MetaFood3D**, which includes 108 mostly consumed food categories with high intra-class diversity, a total of 220 unique food items, each matched to a unique food code, and 637 single food objects in total with each containing nutrition values annotations.

food items within a category. Our fine-grained categorization results in a total of 220 food items, each with a unique FNDDS code, forming the foundation of our 3D data collection process. Including various food items within each category allows our collected 3D models to capture intra-category visual and geometric diversity, enhancing the accuracy of algorithms for dietary assessments. When balancing category diversity against within-category diversity, we chose to prioritize expanding the range of food categories. This decision stems from our belief that generative models have significant potential for data augmentation, enabling scalable expansion of the dataset beyond what manual collection alone can achieve. By focusing on category diversity, our 3D food models can serve as prototypes that can be further enhanced by leveraging internet-scale priors–which would be more challenging if we concentrated solely on within-category variations.

**Data Collection**: We prioritize sourcing real-world food objects from restaurants and ready-to-eat or frozen foods from grocery stores. For food that are difficult to source, we prepare them from raw ingredients such as peanut butter and jelly sandwich. Besides leveraging both the food category and food item categorization, we also enhance intra-class diversity during the data collection step by employing various food sourcing strategies. These include sourcing food from different restaurants, stores, or locations; selecting diverse flavors, brands, breeds, or forms; cutting, peeling, or unwrapping the food; and preparing the food with different ingredients. These strategies ensure that our dataset captures a wide range of appearances and geometries for each food category. Our 3D data collection follows a similar approach to OmniObject3D Wu et al. (2023) and NutritionVerse3D Tai et al. (2023). The food object is placed on a turntable and scanned by a 3D scanner, the Revopoint POP 2[2], which is positioned statically on a tripod. We then record the food's weight and nutrition value. For most objects, keypoint tracking provided by RevoScan software Team (2024b) is sufficient to obtain a 360-degree point-cloud capture of the food object. If the scan is not successful, we manually turn the object. Unlike OmniObject3D Wu et al. (2023), which captures a 360° range, we perform a 720° RGBD video capture by rotating the object twice in a spiral motion, ending with an overhead capture. This approach ensures that we capture the most likely camera angles from typical smartphone users. If the food object can be flipped (e.g., a bowl of beef stew cannot be flipped), we flip the object and repeat the data capture process to capture the underside of non-fluid objects. The depth measurement is obtained using an iPhone App called Record3D Simonik (2023). To ensure precise scale and color measurements, we use calibration fiducial markers Xu et al. (2012) for both camera angle and color calibration. Details of our data collection pipeline can be found in our supplementary materials.

**Annotation**: After collecting the 3D food objects, we perform a series of postprocessing steps and annotate each food object. One of our unique contributions is the annotation of weight and nutrition facts for each food object, which is crucial for food data and dietary assessment tasks. During the data collection process, we record the weight $w_i$ (in grams) of each food object $i$. By leveraging the food code associated with each object, we obtain the nutrient value density $d_i$, which represents the nutrient content per 100 grams of the food item. The nutrient value density is typically expressed as a vector $d_i = [e_i, p_i, c_i, f_i]$, where $e_i$, $p_i$, $c_i$, and $f_i$ denote the energy (in kilocalories), protein (in grams), carbohydrates (in grams), and fat (in grams) per 100 grams of food item $i$, respectively.

---

[2]https://www.revopoint3d.com/pages/face-3d-scanner-pop2

Given the weight $w_i$ and nutrient value density $d_i$, we can accurately determine the total nutrient content $n_i$ for the specific quantity of food object $i$ in our dataset with $n_i = \frac{w_i}{100} \cdot d_i$. The inclusion of weight and nutrition values enables researchers to develop and evaluate algorithms for precise dietary assessment and nutrient estimation. Similarly as in Wu et al. (2023), we also generate data to support various general 3D vision research topics such as point cloud analysis, neural radiance fields, and 3D generation. This includes rendering object-centric and photo-realistic multi-view images using Blender Team (2024a) with accurate camera poses, generating depth and normal maps, and sampling multi-resolution point clouds from each 3D model. Additionally, we provide uniformly sampled video frames with corresponding segmentation masks and depth information. The segmentation masks are generated based on GroundingDINO Liu et al. (2023a), Segment Anything Models (SAM) Kirillov et al. (2023) and Cutie Cheng et al. (2023).

Overall, We collected 637 food objects with 108 food categories. Each food object in our dataset includes the following labels: a scanned 3D object mesh with texture, RGBD video capture of the food both in a standard pose and flipped (if applicable), depth images and masks corresponding to the RGBD video captures, FNDDS food code, nutrition value (energy, protein, carbohydrates, fat), weight value, Blender-rendered frames with normal and depth images, camera parameters used for rendering, and fiducial marker (with known physical dimensions) used in the video capture.

## 4 EXPERIMENTAL RESULTS

In this section, we demonstrate the usage of the MetaFood3D dataset in four downstream tasks: 3D food perception (Section 4.1), novel view synthesis and 3D reconstruction (Section 4.2), 3D food generation and rendering (Section 4.3), and food portion size estimation (Section 4.4). The implementation details of all experiments are available in Supplementary Materials.

| | $OA_{Uniform}$ ↑ | $OA_{Diverse}$ ↑ | $OA_{Clean}$ ↑ | mCE ↓ |
|---|---|---|---|---|
| DGCNN Wang et al. (2019) | 0.862 | 0.198 | 0.725 | 1.000 |
| PointNet Qi et al. (2017a) | 0.822 | 0.179 | 0.672 | 1.210 |
| PointNet++ Qi et al. (2017b) | 0.893 | 0.214 | **0.761** | **0.912** |
| SimpleView Goyal et al. (2021) | **0.919** | 0.219 | 0.747 | 0.992 |
| GDANet Xu et al. (2021b) | 0.903 | 0.206 | 0.740 | 0.935 |
| PAConv Xu et al. (2021a) | 0.892 | 0.199 | 0.711 | 1.036 |
| CurveNet Xiang et al. (2021) | 0.906 | 0.222 | 0.745 | 0.966 |
| RPC Ren et al. (2022) | 0.900 | 0.215 | 0.738 | 0.959 |
| PointMLP Ma et al. (2022) | 0.912 | **0.231** | 0.756 | 1.033 |
| Point-BERT Yu et al. (2022) | 0.914 | 0.226 | 0.729 | 1.013 |

Table 2: **Robustness Analysis** on Intra-class Diversity and Point Clouds Corruption

### 4.1 3D FOOD PERCEPTION

**Intra-class Diversity of Food Shapes**: Food objects in real-world settings are often processed into various shapes, such as whole fruits versus sliced fruits or a single nut compared to multiple nuts in a bowl. To demonstrate the impact of shape diversity on 3D perception algorithms, we select and train 10 existing methods on OmniObject3D and evaluate their performance on both OmniObject3D ($OA_{Uniform}$) and MetaFood3D ($OA_{Diverse}$) using shared food categories. Overall Accuracy (OA) is used to measure the models' robustness against diverse point cloud shapes. Table 2 shows that $OA_{Diverse}$ was generally 70% lower than $OA_{Uniform}$, indicating that models trained with relatively uniform shapes achieved significantly degraded performance on diverse-shaped food test set. This finding highlights the importance of incorporating shape diversity in 3D food datasets, a key strength of MetaFood3D, ensuring the robustness and generalizability of 3D perception algorithms in real-world applications.

**Corruption in Point Clouds**: Real-world 3D point clouds of food items can be affected by various types of corruptions, such as noise, missing points, or scaling issues, arising from factors such as sensor limitations, or variations in scanning conditions. To evaluate the robustness of 3D perception models under these corruptions, we created MetaFood3D-C by modifying MetaFood3D with common corruptions described in Ren et al. (2022). $OA_{Clean}$ represents the overall accuracy on the clean MetaFood3D test dataset. The mean Corruption Error (mCE) Ren et al. (2022) corresponds to

the models tested on the MetaFood3D-C to assess their performance in the presence of real-world corruptions. As shown in Table 2, PointNet++ and GDANet demonstrate the best robustness on average against various corruptions. The full results can be found in the Supplementary Materials.

## 4.2 NOVEL VIEW SYNTHESIS AND 3D MESH RECONSTRUCTION

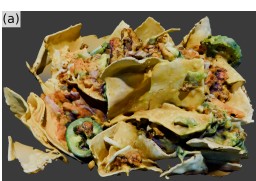 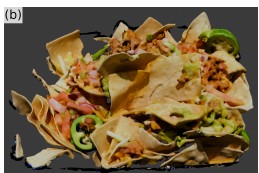 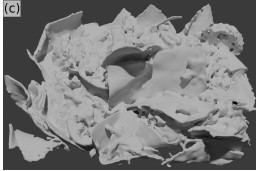 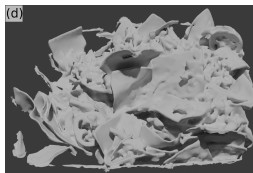

Figure 3: **Reconstructed Mesh: (a)** Ground-truth textured 3D mesh of a complex food item (nachos). **(b)** A textured 3D mesh of the same food item (nachos) reconstructed from video using Nerfacto. **(c)** and **(d)** are mesh-only views of the ground truth and the reconstructed model respectively.

| Method | Input | PSNR (↑) | SSIM (↑) | LPIPS (↓) |
|---|---|---|---|---|
| Nerfacto Tancik et al. (2023) | Render | 19.00 | 0.9162 | 0.0896 |
| | Video | 22.81 | 0.9611 | 0.0718 |
| Nerfacto (masked) Tancik et al. (2023) | Render | 18.87 | 0.9170 | 0.0909 |
| | Video | 9.09 | 0.0557 | 1.0684 |
| Gaussian Splatting Kerbl et al. (2023) | Render | 51.24 | 0.9975 | 0.0044 |
| | Video | 37.75 | 0.9890 | 0.0118 |

Table 3: **Novel view synthesis results** on 108 categories. "Render" represents rendered Blender data from ground truth meshes and "Video" represents captured video data.

In dietary assessment applications, participants are expected to take minimal actions when capturing food-related media, such as recording a short video with limited food pose coverage. These applications serve as ideal test grounds for Novel View Synthesis and 3D Mesh Reconstruction algorithms. In this section, we present preliminary results for these two tasks using both video captures and Blender-rendered images. For novel view synthesis, we select one object per category and apply recent algorithms, Nerfacto Tancik et al. (2023) and Gaussian Splatting (GS) Kerbl et al. (2023), using their official code under default settings. The models are trained on 90% of the data and tested on the remaining 10%. We follow Mildenhall et al. (2021) and report PSNR, SSIM, and LPIPS scores. The results are summarized in Table 3. Upon inspecting the visual results, we observe that Nerfacto struggles with our dataset. In some video-captured scenes, Nerfacto fails to learn the foreground object, resulting in only a pure background color, whereas GS successfully synthesizes all objects. We further tested the Nerfacto method by providing it with foreground masks. Visually, we observed that the foreground was correctly learned, but this approach created artifacts in the background, leading to poor quantitative results as shown in Table 3. Therefore, masking plays a crucial role for the Nerf-based method, Nerfacto, on video data but not on rendered data. This discrepancy highlights the challenging non-uniform sparse views and object scale variations in our video data. For 3D mesh reconstruction, we apply Nerfacto with surface normal prediction settings. Poisson surface reconstruction is then applied to the trained Nerfacto model to obtain the reconstructed mesh. The predicted object meshes from rendered images are compared to the original meshes using Chamfer distance (CD). However, 5 out of 108 objects fail to reconstruct, while the remaining meshes have an average CD of 903.65. For video data, we only provide one of the qualitative results in Figure 3 due to the labor-intensive process of pose alignment with the scanned ground truth object. These results underscore the challenging nature of our dataset.

## 4.3 3D FOOD GENERATION AND RENDERING

Our MetaFood3D dataset enables the generation of highly realistic 3D food objects and facilitates the enrichment of existing 2D food datasets through innovative data synthesis techniques. For 3D object generation, we use GET3D Gao et al. (2022b) to generate textured 3D meshes for various food categories in our dataset. We train the GET3D model from scratch for each selected food type separately, using 3,500 epochs and an average of 750 rendered images per object at a resolution of 512. To compensate for the smaller initial object count compared to the dataset used in GET3D,

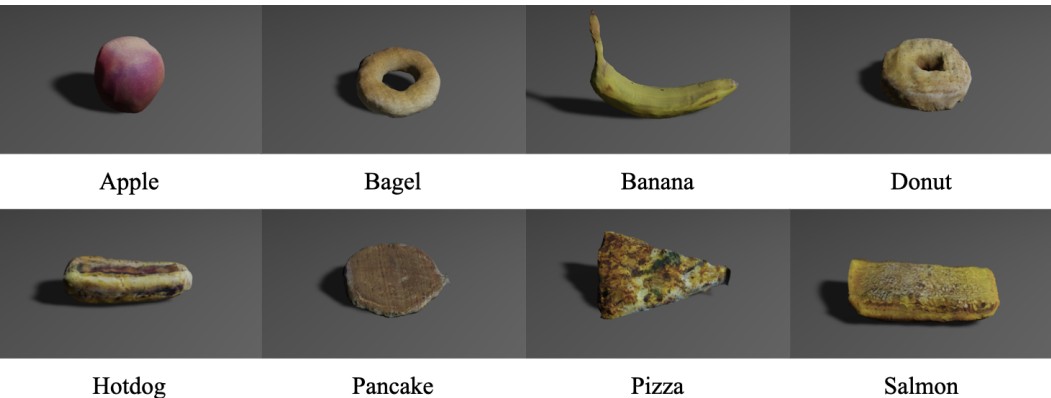

Figure 4: **MetaFood3D utilizes GET3D Gao et al. (2022b) to generate a diverse array of food objects.**

| Food object | Volume (cm$^3$) | Energy Estimate (kCal) | FID ($\downarrow$) | CD x $10^3$ ($\downarrow$) |
|---|---|---|---|---|
| Apple | 278.88 | 217.36 | 105.55 | 5.45 |
| Bagel | 326.04 | 308.58 | 129.01 | 58.48 |
| Banana | 274.69 | 260.01 | 94.26 | 10.75 |
| Donuts | 315.03 | 578.60 | 93.15 | 4.44 |
| Hotdog | 898.01 | 501.44 | 99.81 | 4.69 |
| Pancake | 358.24 | 1205.26 | 106.11 | 42.60 |
| Pizza | 129.83 | 186.37 | 76.09 | 7.10 |
| Salmon | 202.20 | 573.98 | 108.19 | 18.56 |

Table 4: **Qualitative results for different generated food objects** with volume and energy estimates

we set the gamma value to 3,000, penalizing the discriminator and encouraging the generation of more realistic meshes. We demonstrate the quality of the generated objects through FID Heusel et al. (2017) and Chamfer Distance (CD)Barrow et al. (1977) as shown in Table4. A unique aspect of our 3D generation is the inclusion of volume and energy estimates for each generated food object. The energy estimates are calculated based on the generated object's volume, determined using Blender, and the corresponding FNDDS food codes provided by our dataset's nutrition values. This enhances the realism of the generated objects, enables accurate energy calculations, and improves dietary assessment functionalities. Figure 4 visualizes our 3D generation that feature natural textures and coherent shapes enriched by geometric details.

MetaFood3D also allows researchers to generate fully customized 2D synthetic food data by leveraging its collected or generated high-quality 3D food objects, detailed nutrition values, and realistic texture generation capabilities. The dataset supports the creation of synthetic eating scenes with adjustable parameters such as food item placement, portion sizes, and nutrition composition. As shown in Figure 5 (a)(b)(c), we create a breakfast scene in NVIDIA Omniverse simulation engine NVIDIA (2023), complete with ground truth labels such as nutrition values, segmentation masks, and depth map. Additionally, the ground truth of bounding boxes and object 6D poses can also be extracted. These scenes can be automatically generated, as described in Nair et al. (2023). Furthermore, texture generation techniques Chen et al. (2023) can be leveraged to augment food appearances as shown in Figure 5 (d)(e).

### 4.4 FOOD PORTION ESTIMATION

The food portion estimation is a challenging yet important task for food image analysis. Leveraging the rich nutrition value annotations and 3D information in the MetaFood3D dataset, we compare the performance of different portion estimation methods covering the four major approaches (stereo-based, depth-based, model-based, and neural network-based) as discussed in Section 2. Specifically, we sample 2 frames from the captured video for each food item in the dataset. The food items are

Figure 5: **(a) Synthetic scene generation** in NVIDIA Omniverse, composed using individual food objects from MetaFood3D. This scene displays a breakfast plate with associated nutrition values for each item including a total weight of 1,433g, 1,944kCal energy, 70g protein, 103g fat, and 191g carbs. **(b)** Depth map. **(c)** Instance segmentation mask. **(d)** 3D model of an avocado from MetaFood3D, characterized by a brown and dull skin texture. **(e)** The same avocado mesh as in (d), enhanced with a new texture file generated using Text2Tex Chen et al. (2023) with the prompt: *avocado*.

divided into training and testing sets, with one food item per category in the testing set and the remaining items in the training set. Overall, the training set contains 1,036 images, while the testing set consists of 216 images. All methods are evaluated on the same testing set for a fair comparison. We compare the methods using Mean Absolute Error (MAE) and Mean Absolute Percentage Error (MAPE). We use V-MAE and V-MAPE for volume estimation ($cm^3$), and E-MAE and E-MAPE for energy estimation (kCal). Neural network-based methods directly regress energy values, so V-MAE and V-MAPE are not available for them.

| Method | V-MAE | V-MAPE | E-MAE | E-MAPE |
|---|---|---|---|---|
| Baseline | 151.85 | 845.69 | 221.37 | 1287.25 |
| Stereo Reconstruction Dehais et al. (2017) | 135.96 | 210.90 | 271.78 | 244.55 |
| Voxel Reconstruction Fang et al. (2016) | 123.34 | 104.07 | 190.38 | 145.31 |
| RGB Only (ResNet50) Shao et al. (2021) | - | - | 1932.01 | 1124.9 |
| Density Map Only (ResNet50) Vinod et al. (2022) | - | - | 1100.39 | 663.43 |
| Density Map Summing Ma et al. (2023) | - | - | 436.12 | 142.44 |
| 3D Assisted Portion Estimation Vinod et al. (2024) | 195.92 | **79.33** | 260.79 | **102.25** |

Table 5: **Comparison of image-based dietary assessment methods on the MetaFood3D dataset.**

The results presented in Table 5 demonstrate the performance of different classes of existing methods on our MetaFood3D dataset. The 3D Assisted Portion Estimation method Vinod et al. (2024) achieves the lowest V-MAPE and E-MAPE. However, it is important to note that this method has higher V-MAE and E-MAE values compared to some other approaches, which suggests that the incorporation of 3D object information can lead to improved performance in terms of percentage errors, but further research may be needed to reduce the absolute errors. The MetaFood3D dataset provides a valuable resource for developing and evaluating various dietary assessment techniques, and the presented results highlight the potential for further improvements in food portion estimation accuracy.

## 5 CONCLUSION

In this paper, we present MetaFood3D, a food-specific 3D object dataset to advance dietary analysis. This new dataset provides a robust benchmark for developing and evaluating 3D vision algorithms for real-world scenarios. The dataset features diverse intra-class variations, detailed nutrition annotations and rich multimodal data. Experimental results demonstrate great potential for using our dataset in various downstream tasks related to food image analysis.

**Limitations:** Our food list is selected from the American diet, thus it may not accurately represent the diversity of diets in other regions of the world.

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

## APPENDIX

This section provides comprehensive information on our MetaFood3D datasets and detailed implementation specifications for our experiments. Due to the large file size of the complete dataset, we have included only the 3D point clouds of all 637 food items and corresponding nutrition values in the supplementary zip file. The point clouds provided are randomly sampled from the mesh with 1024 points and 4096 points. The full dataset, including all annotations, will be made publicly available.

## A    DATASET INFORMATION

**Data distribution:** In this supplementary material, we present a comprehensive distribution figure (Figure 7) that includes the names of all categories.

**Intended uses:** In experiments section of the main paper, we showcase the intended uses of our dataset. These include 3D food perception, novel view synthesis, 3D mesh reconstruction, 3D food object generation, synthetic food intake scene image and data generation, 3D food object texture augmentation, and food portion estimation. The dataset is created to facilitate tasks and downstream applications in both the dietary assessment domain and the 3D vision domain.

**Example nutrition values and video captures**: Due to space constraints in the main paper, we provide an example here showcasing the nutritional values associated with our dataset in Figure 6. Additionally, we present examples of our video captures and the provided object masks, as illustrated in Figure 8.

| Food_Type | Object_Name | FNDDS Food Code |
|---|---|---|
| asparagus | new_asparagus_1 | 75202027 |
| **Weight (g)** | **Energy (kcal)** | **Protein (g)** |
| 4 | 1.96 | 0.0916 |
| **Fat (g)** | **Carbs (g)** | **Volume** |
| 0.13 | 0.1616 | 5.11 |
| **Main food description** | | |
| Asparagus, fresh, cooked with oil | | |

Figure 6: Nutritional information of a food sample from MetaFood3D

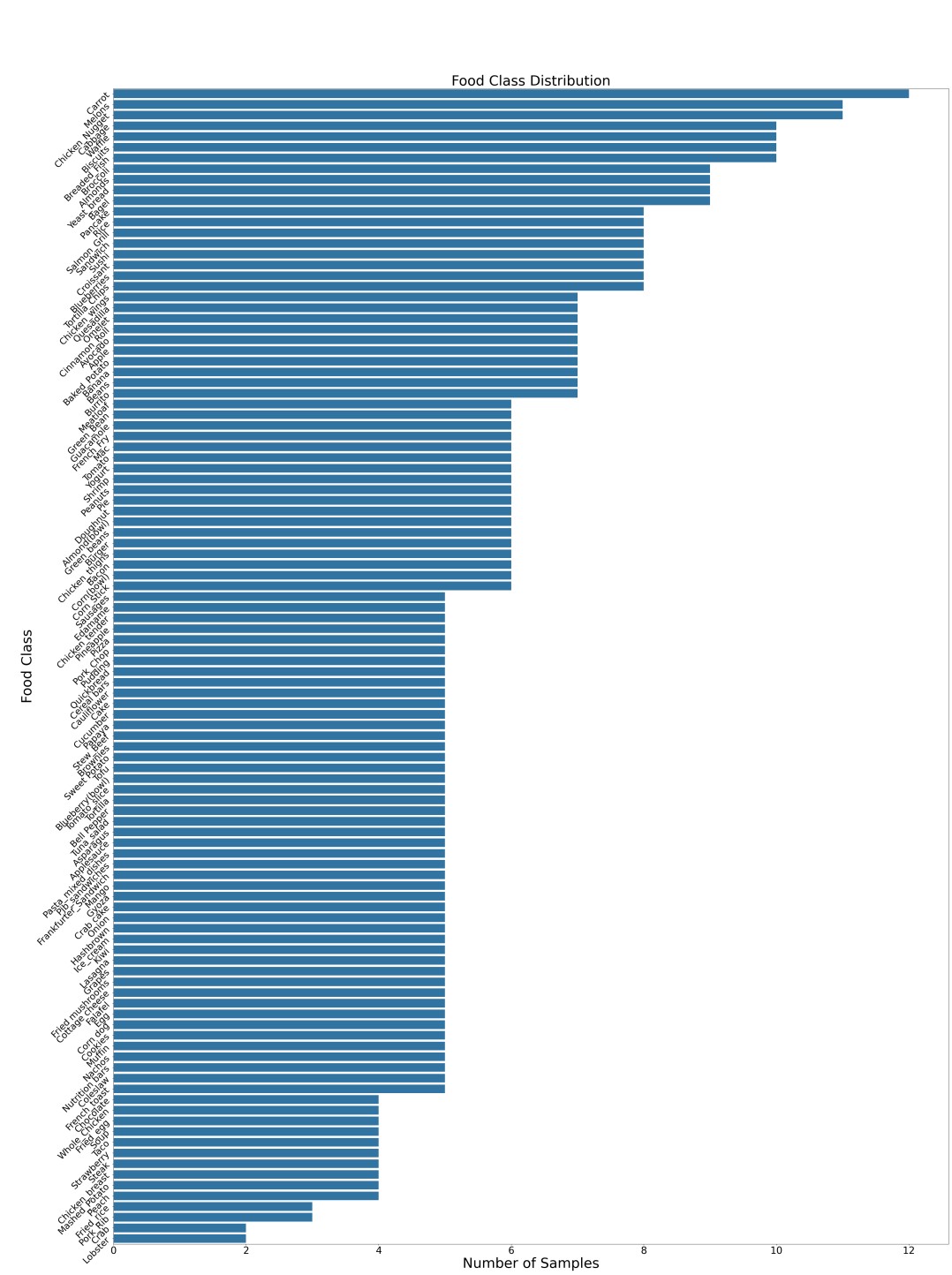

Figure 7: MetaFood3D Overview: Complete Distribution of Food Samples by Class.

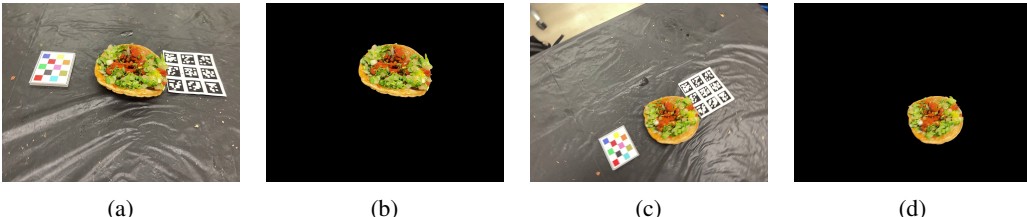

|       |       |       |       |
| :---: | :---: | :---: | :---: |
| (a)   | (b)   | (c)   | (d)   |

Figure 8: Example video capture frames with their masks. (a), (c) two example frames for the food object taco, showing different camera views of the same object. (b), (d) results after applying the provided segmentation masks.

## B    DATA COLLECTION DETAILS

In this section, we provide additional details about our data collection pipeline, illustrated in Figure 9. Each food object is captured/measured using a scanner, an iPhone app, and a scale. The resulting data are then consolidated and uploaded to our cloud storage. Metadata, along with any potential capture issues, is entered manually. These issues are addressed during the post-processing stage. For RGBD video capture, we employ a 720° approach by rotating the object twice in a spiral motion, concluding with an overhead capture. Figure 11 illustrates an example of our video capture camera trajectory computed using COLMAP Schönberger & Frahm (2016). As depicted, our camera movements are varied and noisy, effectively reflecting real-world capture scenarios. To ensure precise scale and color measurements of the video capture, we include calibration fiducial markers Xu et al. (2012) in the video, as shown in Figures 10(a) and 10(b). The dimensions and colors of the markers are provided in the dataset. Additionally, the POP2 scanner required a dark-colored background for accurate capture. Using a brighter background resulted in parts of the background being erroneously captured by the software as part of the model. Therefore, the turntable and video capture setup were covered with a non-reflective disposable black liner, as shown in Figures 10(c) and 10(d).

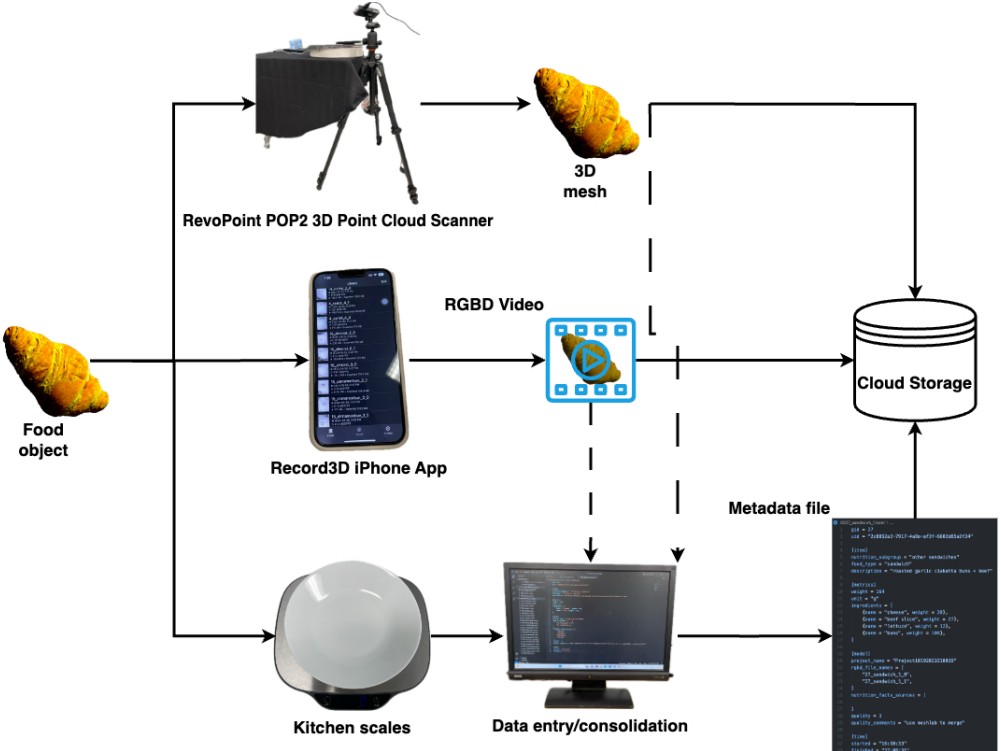

Figure 9: Data collection pipeline used for the MetaFood3D dataset.

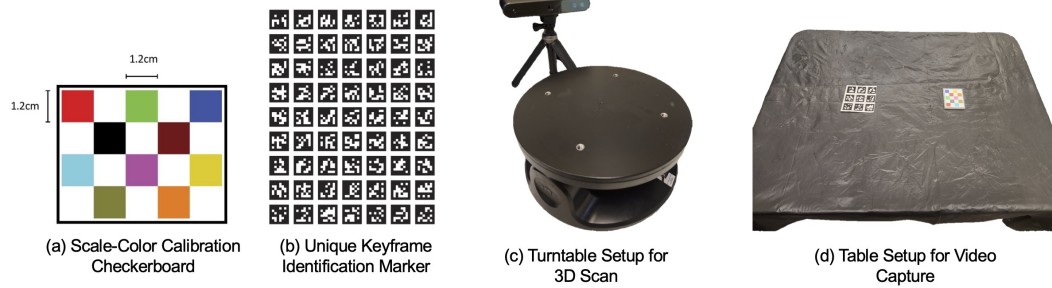

(a) Scale-Color Calibration Checkerboard

(b) Unique Keyframe Identification Marker

(c) Turntable Setup for 3D Scan

(d) Table Setup for Video Capture

Figure 10: Data capture physical setup elements.

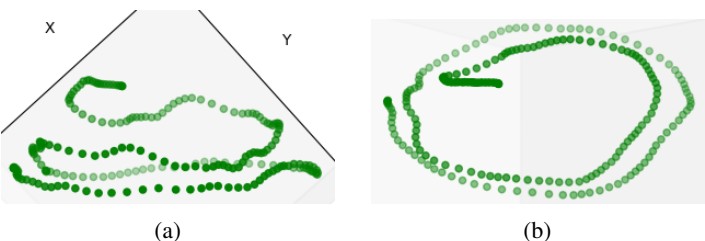

(a)

(b)

Figure 11: Example camera centers for video capture. Our setup involves a 720-degree capture of the food object from various camera angles, concluding with a top-down view. (a) Side view of the camera trajectory. (b) Top view of the camera trajectory.

**Challenges in data collection:** Collection food object data is relatively more challenging than collecting rigid object data. We list the challenges in the following.

*Lighting:* We encountered difficulties in capturing intensely red objects, such as strawberries or tomatoes, under cool lighting. In these cases, the automatic object tracking in the RevoScan software would fail to follow the object on the turntable. We resolved this issue by switching to a warmer light source positioned directly above the turntable.

*Food container:* Certain food items imply ambiguitiy in the method of preparation. For example blueberries could both be depicted as singular berries as well as as a container full of berries, the latter being more representative of what is typically visible in a food scene. For this reason some of the objects in our dataset do contain dishes in view and they are annotated accordingly. Moreover, some of the food items are fluid or liquid in nature and capturing them without a container would not be physically possible.

*Object perspective inconsistencies:* Certain composite food items such as sandwiches have proven exceptionally problematic. Loose elements of the stuffing in a sandwich, for example salad or ham, upon inversion naturally bend in the opposite direction due to gravity. To address this the flexible parts have been edited out in the inverted view of the mesh.

*Object sizes:* Thin and small objects have yielded few problems. It was more problematic to accurately capture large, oblong objects such as bananas. To ensure good texture quality and point-cloud density the scanner should remain relatively close to the object. For oblong objects the distance to the camera must be increased, as the object must stay within the frame for successful capture. This can be in fact addressed by careful positioning of the item on the turntable and using the shortest possible distance to the camera, ensuring the object is fully visible in the frame. Capturing such objects required multiple trials and was more time consuming than an average object capture.

*Reflective objects:* We discovered that shiny or reflective objects posed significant challenges during scanning. The device struggled to synchronize points on these objects due to light reflections. Consequently, we were unable to scan items such as cherries in light syrup or popsicles. These objects have been excluded from our dataset.

*Pile-shaped objects:* Some of the food items we captured were largely unstructured piles, such as shredded carrots. For these objects, we had to take extra care to move them undisturbed from the RGBD capture location to the turntable. To facilitate this transfer, we used a small rectangular piece of black liner, and two individuals were tasked with carefully moving the pile.

**Data collection time:** We source our food from restaurants, grocery stores, or by preparing it from scratch. Some food items can be purchased together, but due to perishability, we only purchase small batches at a time. On average, it takes about 10 minutes to source each item. Each food object requires approximately 20 minutes of work by two people for data collection. The average time spent on post-processing each model is about 40 minutes. In total, each food object requires an average of 1.5 hour of person-hours.

## C    IMPLEMENTATION DETAILS OF THE EXPERIMENTS

In this section, we provide implementation details that were omitted from the experiment section of the main paper due to space constraints.

### C.1    3D FOOD PERCEPTION

In Table 6, we present the robustness analysis of ten 3D point cloud classification models on corrupted point clouds, including DGCNN Wang et al. (2019) as the baseline. The clean point clouds in MetaFood3D are sampled from ground truth mesh files, making them highly accurate representations of the actual physical models' shapes and true dimensions. However, when using point clouds in practical applications, such as 3D reconstructed point clouds from videos, the resulting point clouds often include distorted scaling, coordinate jitters, or changes in the number of points. Therefore, it is crucial to evaluate the performance of point cloud networks under standardized corrupted test sets, as shown in Figure 12.

MetaFood3D-C test sets are generated with seven types of corruptions: Scale, Rotate, Jitter, Add-G, Add-L, Drop-G, and Drop-L following the standard pipeline in Ren et al. (2022). All models were trained on the clean MetaFood3D train dataset. The overall accuracy on the clean MetaFood3D test set is denoted as $OA_{Clean}$. The calculation for Corruption Error(CE)1 and mCE2 are consistent with those in Ren et al. (2022):

$$CE_i = \frac{\sum_{l=1}^{5}(1 - OA_{i,l})}{\sum_{l=1}^{5}(1 - OA_{i,l}^{\mathrm{DGCNN}})} \tag{1}$$

$$mCE = \frac{1}{N}\sum_{i=1}^{N} CE_i \tag{2}$$

We observed that compared to the baseline model DGCNN, many models exhibit strong robustness to the Drop-G corruption method, while most models show poor robustness to the Add-G corruption method. Additionally, we can see that Point-BERT, which employs Bert-style pretraining, performs exceptionally well under the Scale corruption. On average, PointNet++ and GDANet demonstrate the best robustness to point cloud corruption.

**Training/Testing Settings**: From left to right, for columns 1 and 2 of Table 2 in the main paper, all models are trained on the OmniObject3DWu et al. (2023) training set. $OA_{Uniform}$ represents the performance of these models evaluated on the OmniObject3D test set. $OA_{Diverse}$ represents the performance of these models evaluated on the MetaFood3D test set.

For columns 3 and 4 of Table 2 in the main paper and for the full Table 6, all models are trained on the MetaFood3D training set. $OA_{clean}$ represents the performance of these models evaluated on the MetaFood3D test set. mCE is calculated based on the performance of these models evaluated on MetaFood3D-C, which is generated with the MetaFood3D test set corrupted by different methods and degrees.

OmniObject3D and MetaFood3D do not have official training/test set splits. Therefore, we used a random split with a ratio of 8:2 for training and testing samples from the same category.

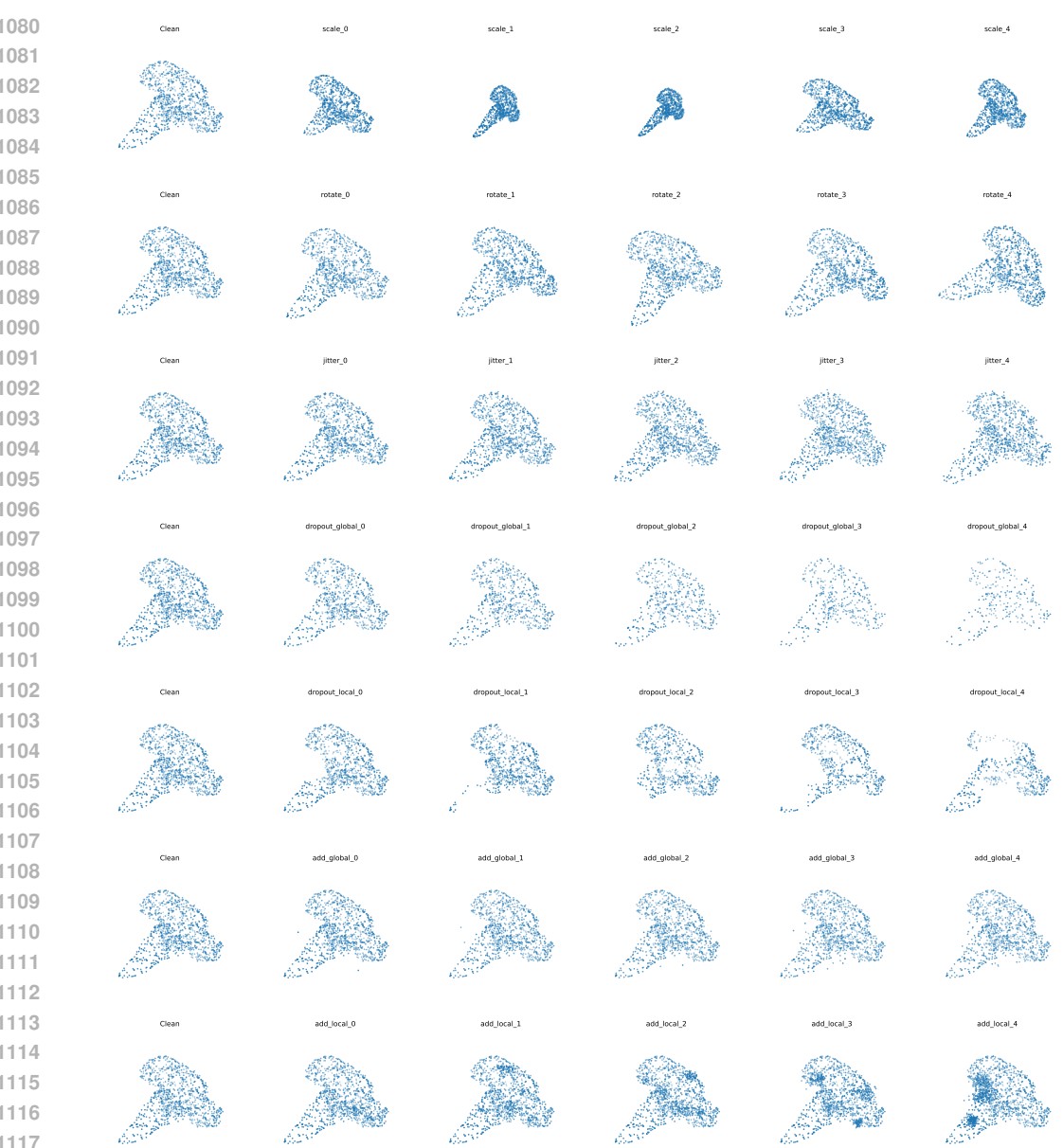

Figure 12: MetaFood3D-C Example: This image illustrates the corruptions of a broccoli. Each row represents a different corruption method, with the degree of corruption increasing from left to right.

**Compute Resources**: All 3D point cloud perception models were trained on a single NVIDIA A40. Except for PointNet++, all models training were finished in 3 hours, while PointNet++ took approximately 5 hours. The GPU memory usage for all 3D point cloud perception models was within 5000MB. The learning rate, optimizer, and other hyperparameters were set according to the official repositories of each model. Please refer to the "License information for code used" section for details.

## C.2 NOVEL VIEW SYNTHESIS

We used NerfStudio Tancik et al. (2023) for experiments on Nerfacto Tancik et al. (2023) and the Gaussian Splatting Kerbl et al. (2023) official repository for experiments on Gaussian Splatting.

| | OA$_{\text{Clean}}$ ↑ ‖ | Scale | Jitter | Drop-G | Drop-L | Add-G | Add-L | Rotate | mCE ↓ |
|---|---|---|---|---|---|---|---|---|---|
| DGCNN Wang et al. (2019) | 0.725 | 1.000 | 1.000 | 1.000 | 1.000 | 1.000 | 1.000 | 1.000 | 1.000 |
| PointNet Qi et al. (2017a) | 0.672 | 1.274 | 0.900 | 0.889 | 1.050 | 1.762 | 1.219 | 1.374 | 1.210 |
| PointNet++ Qi et al. (2017b) | **0.761** | 1.027 | 0.950 | **0.732** | 0.895 | **0.832** | **0.752** | 1.197 | **0.912** |
| SimpleView Goyal et al. (2021) | 0.747 | 0.917 | 1.011 | 0.924 | 1.011 | 1.021 | 1.083 | 0.976 | 0.992 |
| GDANet Xu et al. (2021b) | 0.740 | 0.908 | 0.951 | 0.946 | 0.993 | 0.901 | 0.931 | **0.913** | 0.935 |
| PAConv Xu et al. (2021a) | 0.711 | 1.051 | 1.014 | 0.971 | 1.013 | 1.163 | 1.027 | 1.015 | 1.036 |
| CurveNet Xiang et al. (2021) | 0.745 | 0.989 | **0.807** | 0.855 | 1.005 | 1.160 | 1.028 | 0.919 | 0.966 |
| RPC Ren et al. (2022) | 0.738 | 0.969 | 0.985 | 0.990 | **0.860** | 0.940 | 0.941 | 1.031 | 0.959 |
| PointMLP Ma et al. (2022) | 0.756 | 0.982 | 1.056 | 0.897 | 0.984 | 1.294 | 1.077 | 0.942 | 1.033 |
| Point-BERT Yu et al. (2022) | 0.729 | **0.707** | 1.006 | 0.919 | 0.962 | 1.334 | 1.101 | 1.059 | 1.013 |

Table 6: Robustness Analysis on Corrupted Point Clouds

**Datasets**. Throughout, we used train-test split of 9:1. We randomly split the dataset into train set and test set. The split is recorded and used for all the experiments. For video input, we applied COLMAP for extracting the camera parameters and applied the data processing pipeline from NerfStudio to prepare our datasets. For blender input, we used blender-rendered frames as described in the Annotation paragraph in Section 3 and prepared our dataset in the NerfStudio blender data format.

**Training details**. All Nerfacto Tancik et al. (2023) and Gaussian Splatting Kerbl et al. (2023) models were trained for 30,000 iterations before evaluation. For Nerfacto training, we used the default learning rate of 0.0005 as defined in the NerfStudio official repository. For Gaussian Splatting training, we used the following default values: position lr of 0.00016, feature lr of 0.0025, opacity lr of 0.05, scaling lr of 0.005, and rotation lr of 0.001.

**Compute resource**. We used a pool of NVIDIA RTX 6000 Ada Generation, NVIDIA GeForce RTX 4090, and NVIDIA RTX A6000 GPUs. As training was done frame-by-frame, GPU memory consumption was low and lower-tier GPUs could be used to reproduce our results. Namely, memory consumption was 3367 MiB for Nerfacto unmasked training, 3565 MiB for Nerfacto masked training, and 4295 MiB for Gaussian Splatting.

## C.3 3D Food Generation

For our experiments, we utilized GET3D Gao et al. (2022b) for 3D generation, following the procedures outlined in the official repository.

**Image rendering**: For 3D generation, we rendered 1,500 images for each subcategory of food models from our dataset using Blender. For example, if the food model "Apple" consists of 5 different kinds of Apple models, we generated 7,500 images for the Apple category. These images were captured from various angles at a resolution of 512 and included camera parameters such as elevation and rotation.

**Training details**: Each model was trained individually for 3,500 iterations. A gamma value of 3,000 was used to heavily penalize the discriminator, ensuring accurate representation of the food objects, particularly given the smaller number of sub-models per object category. The training process, using 4 GPUs, took 1.5 days to complete 3,500 iterations, achieving a presentable FID score.

**Evaluation Metrics**: To evaluate the geometry we use **Chamfer Distance** (CD) to measure the similarity between two sets of points in 3D space. Let $X \in S_g$ denote a generated shape and $Y \in S_r$ a input reference. To compute $CD$, we first randomly sample $N = 2048$ points $X_p \in \mathbb{R}^{N \times 3}$ and $Y_p \in \mathbb{R}^{N \times 3}$ from the surface of the shapes $X$ and $Y$, respectively Gao et al. (2022b). The $CD$ can then be computed as:

$$CD(X_p, Y_p) = \sum_{\mathbf{x} \in X_p} \min_{\mathbf{y} \in Y_p} \|\mathbf{x} - \mathbf{y}\|_2^2 + \sum_{\mathbf{y} \in Y_p} \min_{\mathbf{x} \in X_p} \|\mathbf{x} - \mathbf{y}\|_2^2. \tag{3}$$

To assess the quality of the generated textures and geometry, we use the **Fréchet Inception Distance** (FID) metric. Following the implementation in GET3D Gao et al. (2022b), we render 50,000 views of the generated shapes for each category using camera poses randomly sampled from a predefined distribution. All images in the test set are encoded using a pretrained Inception v3 model Szegedy

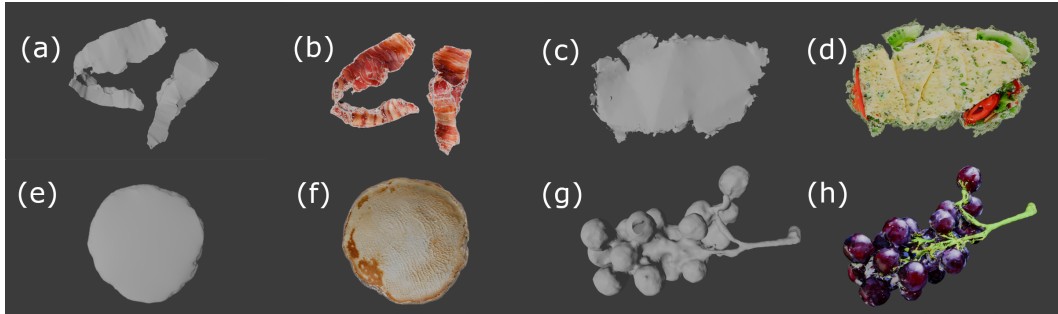

Figure 13: Sample textures generated by Text2Tex. (a, b) Bacon mesh vs. generated texture. (c, d) Omelette mesh vs. generated texture. (e, f) Pancake mesh vs. generated texture. (g, h) Grapes mesh vs. generated texture.

et al. (2016), with the output from the last pooling layer used as the final encoding. The FID metric is then calculated as follows:

$$\text{FID}(S_g, S_r) = \|\mu_g - \mu_r\|_2^2 + \text{Tr}[\Sigma_g + \Sigma_r - 2(\Sigma_g \Sigma_r)^{1/2}] \tag{4}$$

where $\mu_g$ and $\Sigma_g$ are the mean vector and covariance matrix of the generated image encoding, and $\mu_r$ and $\Sigma_r$ are the mean vector and covariance matrix of the encoding from the input images. Tr denotes the trace operation.

**Compute resources**: We utilize 4 NVIDIA A40 GPUs for the 3D generation task. Training each food model individually over 3500 iterations takes approximately 1.5 days. During GET3D training, memory consumption was 33,600 MiB per GPU.

## C.4 RENDERING

**Texture rendering:** In the main paper, we provided an example of using a texture generation model, Text2Tex Chen et al. (2023), to change the appearance of a food object. In this section, we present additional qualitative samples, as shown in Figure 13.

**Parameters and Compute resources**: All example samples were generated on an NVIDIA RTX A6000 GPU. The prompt provided consisted of the category name of the object. Text2Tex was run with default parameters: 20 update steps, 50 DDIM generation steps, an update strength of 0.3, a view threshold of 0.1, and 36 viewpoints. Automated post-processing of the texture was also enabled.

## C.5 PORTION ESTIMATION

The metrics used for portion estimation method comparison are:

$$\mathbf{MAE} = \frac{1}{N} \sum_{i=1}^{N} |(\hat{v}_i - v_i)| \tag{5}$$

$$\mathbf{MAPE\ (\%)} = \frac{1}{N} \sum_{i=1}^{N} \frac{|(\hat{v}_i - v_i)|}{v}$$

where $v_i$ is the ground-truth value, $\hat{v}_i$ is the estimated value of the $i$-th image, and $N$ is the number of images in the dataset. The main takeaway is the usability of the MetaFood3D dataset for the different input requirements posed by the portion estimation methods. We compare across different classes of methods described in Section 2:

**Baseline** A model that always predicts the mean value of the field is used. The error between each instance in the dataset and the mean of the dataset is established in this baseline. For the Volume MAE (V-MAE) and Volume MAPE (V-MAPE), the mean volume of the dataset is always predicted while for the Energy MAE (E-MAE) and Energy MAPE (E-MAPE) the mean energy of the dataset is always predicted.

**Stereo Based Methods** The stereo reconstruction method in Dehais et al. (2017) describes a process for using 2 images for keypoint detection and matching, stereo rectification, disparity map, and depth map calculation. However, since there is no publicly available implementation, we use the pipeline described in Dehais et al. (2017) but replace the feature-matching framework with LightGlue Lindenberger et al. (2023) for better results. The disparity map along with some camera epipolar geometry information is used to project the points to 3D space to obtain a point cloud. The ground-truth segmentation map is used to filter the points to have only the foreground. The volume is scaled using the mean point-to-volume ratio on the training split. This scale is different for each food type. Finally, the volume scaling is used on the reconstructed point clouds on the test dataset to obtain the volume estimate.

**Depth Based Methods** A depth based reconstruction method described in Fang et al. (2016) is implemented with the depth map in the MetaFood3D dataset. The depth map is decoded to actual values using the conversion process detailed by the depth capture mobile app. The RGB image is converted to HSV, the luminosity value scaled from 0 to 3 encodes the depth information in meters. This converted depth map is then used to create a point cloud representation of the scene. The same process applied for the stereo reconstruction is used to scale the point cloud to the actual volume using the point-to-volume ratio of the training dataset. Finally, we obtain the energy from the estimated volume using the same scaling used before.

**Neural Network Based Methods** Three neural network based methods are implemented, RGB Only (Resnet50) Shao et al. (2021), Density Map Only (ResNet50) Vinod et al. (2022) and Density Map Summing Ma et al. (2023). The neural network based methods are trained to estimate the food energy directly and hence do not have any intermediate volume estimates. In RGB Only (Resnet50) Shao et al. (2021) the RGB image serves as an input to a network with a ResNet50 He et al. (2016) backbone feature extractor. The extracted features are then fed to some linear layers with the final linear layer having 1 output which is the estimated energy. The network is supervised on the L1 Loss between the ground-truth energy and estimated energy. For the other methods, we implement the concept of an energy density map. Here, we utilize the ground-truth segmentation maps to understand the area occupied by the foods in the image. Then, the ground-truth energy of the food is distributed uniformly over this area and then scaled to have the pixels maximum value as 255 over the whole dataset. This "Energy Density Map" now contains information about the energy of the food. We use the ground-truth energy density map directly via a Resnet50 He et al. (2016) feature extractor, a few linear layers for estimating the energy. Finally, the Density Map Summing Ma et al. (2023) method utilizes this "Energy Density Map" and sums up the values of all the pixels and scales it based on the factor used to create the maps. The only error introduced in this approach is the quantization loss resulting from conversion of the "Energy Density Maps" to images. For the later couple of methods, the ground-truth "Energy Density Map" is used although the original implementation uses a generative model to learn this mapping. However, our implementation should yield better results because the ground-truth maps are used directly. This is done since the ground-truth volume scaling is utilized in the reconstruction methods. Therefore, in order to maintain a fair comparison, the ground truth is utilized directly.

**Model Based Methods** The 3D Assisted Portion Estimation Vinod et al. (2024) utilizes the 3D model of the food to estimate the volume through image rendering and model scaling. For this method, the checkerboard pattern in the image is used to estimate the orientation and translation of the camera and the object in 3D space. Therefore, for the images where the marker was not automatically detected, we manually annotated the corner points as input to the method. Further, only the testing images were used for evaluating the method but the 3D models for each food type were taken from the training dataset. This means that none of the 3D models for any of the images in the testing dataset were used for evaluation. To keep it fair with the other methods, the ground-truth segmentation maps were used. Despite this, the best performance over the multiple methods were shown for the 3D Assisted method which demonstrates the generalizability and the power of the 3D models in portion estimation.

