# OpenReview forum: "MetaFood3D: 3D Food Dataset with Nutrition Values"
_ICLR.cc/2025/Conference — ICLR 2025 Conference Withdrawn Submission_

### Official Review · Reviewer_k8gP · 2024-10-31

**Soundness:** 3
**Presentation:** 3
**Contribution:** 2
**Rating:** 5
**Confidence:** 2

**Summary:**

This paper proposes a dataset for food in particular. Given the rich type of annotations the dataset provides, various experiments/models are benchmarked on the dataset.

**Strengths:**

- A multi-modal dataset on food categories is important in the relevant community.
- According to Table 1, the proposed dataset seems to include more types of annotations than any prior datasets.
- Presentation is mostly clear. Important stats are provided as well.
- A major contribution of this paper is its thorough evaluation of the proposed dataset, with a total number of 4 different tasks.

**Weaknesses:**

- My major concern is the scale of the proposed dataset. Although there are 108 different food categories, there are only 637 samples. Compared to OmniObject3D which does not provide nutrition information, the proposed dataset yields only ~20 more categories but with <1/4 number of samples. This limits the contribution and future potential of the dataset --- it can only be used for evaluation not for training.
- There is no new method proposed along with the dataset, which weakens the contribution further. Again, this could be due to the reason that the dataset can not be used for training, which limits its use case and prevents specialized methods from being proposed.
- Some of the annotations are not really ground truth. I could be wrong, but according to my understanding, annotated depth maps are obtained via blender, which corresponds to the reconstructed 3D model rather than the actual depth captured by some sensor. Moreover, segmentation masks seem to be inferred from off-the-shelf models directly rather than being manually labeled. These raise concerns about the accuracy of the annotations.

**Questions:**

I like the direction of this paper and I agree with how important a food dataset can be. However, I think there are more work needs to be done to make the dataset larger scale and/or with more accurate annotations.

---

### Official Review · Reviewer_dmoG · 2024-11-03

**Soundness:** 3
**Presentation:** 3
**Contribution:** 3
**Rating:** 5
**Confidence:** 4

**Summary:**

This paper introduces a new dataset MetaFood3D, which aims to advance the dietary analysis from 2D to 3D. It includes 637 3D food objects across 108 categories with rich modalities, such as textured mesh files, RGBD videos, and segmentation masks. This paper demonstrates four downstream tasks: 3D food perception, novel view synthesis and 3D reconstruction, 3D food generation and rendering, and food portion size estimation.

**Strengths:**

1. It is interesting to establish a 3D dataset for food, which also provides different sources, such as textured meshes, RGB-D videos, and segmentation masks.
2. It provides multiple downstream tasks, including 3D food perception, novel view synthesis, 3D mesh reconstruction, 3D food generation, and food portion estimation.

**Weaknesses:**

1. The food list is derived from the American diet, which may limit its applicability and representation of dietary diversity in other regions.
2. Compared to existing datasets, this dataset seems to be not large enough in scale, and its design does not sufficiently highlight the characteristics of the food category.

**Questions:**

1. How does the dataset handle seasonal variations in food items, which could affect the accuracy of dietary assessments over different periods?
2. What measures are in place to ensure the long-term maintenance and updating of the dataset to reflect changes in food types and nutritional values?
3. Could the authors elaborate on the potential of MetaFood3D for applications beyond dietary assessment, such as in the food industry or culinary arts?
4. The authors referenced classic Food-101 dataset in “Related Work”, why there is no comparison in Table 1?
5. In figure 2, the images shown on the 3D Food Data Distribution have no correspondence with the food names on the abscissa. Please clarify it.
6. What’s the difference among the objects in the same category? Is there a standardization to select them? I checked the supplementary files, for example, I didn’t see much difference between point clouds in “Apple” category.
7. For the can-be-flipped objects in section 3 “Data Collection”, since they have been captured twice, will it be two meshes provided? Or is there a way to integrate both captured data to make a better 3D mesh?
8. In table 3, the experiment with masks shows poorer performance, the authors did explain the reason, but they also mentioned “masking plays a crucial role for the Nerf-based method”. To support this point, it is better to calculate the metric with rendered results with masks to focus on the foreground.
9. There is no visual results about 3DGS, please add them, as well as more Nerfacto’s results.
10. In section 4.2, why there is no evaluation on 2DGS, since authors have already mentioned it in “Related Work”?
11. In section 4.2, except for Poisson surface reconstruction, I think it is worth to try nerf2mesh [Delicate Textured Mesh Recovery from NeRF via Adaptive Surface Refinement.] to reconstruct 3D mesh.
12. In section 4.2, why there is no mesh reconstruction based on gaussian splatting’s results? As I know, there are some algorithms such as 2DGS and SuGaR [Surface-Aligned Gaussian Splatting for Efficient 3D Mesh Reconstruction and High-Quality Mesh Rendering] could do the reconstruction work.
13. For the “volume” mentioned in table 4, can you give a detailed description on how to calculate it?
14. The authors presented more detailed food code representation to enable more precise dietary analysis. However, there is no comparative experiments between previously used food code and new code in the paper, to make the conclusion more solid.
15. For “challenges in data collection” in the appendix, it is better to provide visual examples with textual description.

---

### Official Review · Reviewer_4vvt · 2024-11-03

**Soundness:** 3
**Presentation:** 3
**Contribution:** 2
**Rating:** 5
**Confidence:** 3

**Summary:**

The paper introduces MetaFood3D dataset to aid in dietary assessment and computer vision tasks. This dataset consists of 637 3D-scanned food items across 108 categories, each annotated with nutritional values, weight, and food codes together 3D meshes, segmentation masks, depth maps, etc. Experiments show that the proposed dataset supports different downstream tasks, including 3D food perception, novel view synthesis, 3D food generation, and portion estimation.

**Strengths:**

The dataset fills a gap in the field by providing 3D food data with detailed nutritional annotations. The authors benchmark different tasks using the proposed dataset.

**Weaknesses:**

- The food categories reflect only American dietary patterns, which may limit the dataset’s applicability for global diets.

- The dataset includes only 637 objects, which may restrict the generalizability of models trained on it, especially for categories with high visual variability.

- The novelty is limited, which is not enough for ICLR conference. The authors only propose the dataset and benchmarks using existing methods. There is no contribution about proposing methods.

**Questions:**

- Can we trust the information about calories annotated in the dataset?

---

### Official Review · Reviewer_2mbP · 2024-11-04

**Soundness:** 3
**Presentation:** 3
**Contribution:** 3
**Rating:** 6
**Confidence:** 4

**Summary:**

This work presents MetaFood3D, a dataset of 3D scans of real-world food items. The dataset consists of about 600 food samples from 108 categories, with multi-view images, depth, instance masks, meshes, size calibration and nutrition information. Datasets such as this are important for developing and evaluating computer vision systems for important capabilities such as nutrition estimation and portion size estimation. It is the largest dataset to date of its kind.

The paper includes empirical investigations on 3D recognition, 3D shape reconstruction, 3D food generation/rendering and food portion estimation, showcasing the utility of the dataset. The proposed dataset is an important contribution enabling future work in this domain.

**Strengths:**

- The dataset covers all important aspects of a dataset that can be used for estimating properties like portion size or nutrition content: multi-view images, instance masks, size and nutrition information.
- The most important distinction to prior work is the high diversity and large scale while also including 3D information. This is important for nutrition estimation and portion size estimation, as these applications require absolute measurements.
- The evaluations of what the dataset can be used for are thorough, covering 3D recognition, 3D reconstruction, 3D generation and food image/scene rendering, and portion size estimation.
- For the task of portion size estimation, empirical evidence is presented that having access to 3D information can improve performance
- The paper is well written and easy to follow, and the decisions behind the data collection process are explained and justified well.

**Weaknesses:**

- The paper would be greatly improved if in addition to a dataset, there was a well designed benchmark specifically built around applications like nutrition or portion size estimation, including in the wild testing data collected with various smartphones in various contexts. This would tie it more closely
- The application of "food generation" seems a bit arbitrary. It would be very helpful if there was some explanation//justification as to why this is useful and what is the impact of being able to generate high quality 3D food assets.


Minor:
- Citation typos on L279 and 280

**Questions:**

Table 4 is included but doesn't have a discussion in the main text. What do the numbers shown in the table mean?
It would be great if the authors can discuss the main points that were raised in the weaknesses section.

---

### Note · Authors · 2024-11-14

I have read and agree with the venue's withdrawal policy on behalf of myself and my co-authors.